# Effectiveness and cost-effectiveness of a cardiovascular risk prediction algorithm for people with severe mental illness (PRIMROSE)

Ella Zomer,[1,2] David Osborn,[3,4] Irwin Nazareth,[2] Ruth Blackburn,[3] Alexandra Burton,[3] Sarah Hardoon,[2] Richard Ian Gregory Holt,[5] Michael King,[3] Louise Marston,[2] Stephen Morris,[6] Rumana Omar,[7] Irene Petersen,[2] Kate Walters,[2] Rachael Maree Hunter[2]

## ABSTRACT

**Objectives** To determine the cost-effectiveness of two bespoke severe mental illness (SMI)-specific risk algorithms compared with standard risk algorithms for primary cardiovascular disease (CVD) prevention in those with SMI.

**Setting** Primary care setting in the UK. The analysis was from the National Health Service perspective.

**Participants** 1000 individuals with SMI from The Health Improvement Network Database, aged 30–74 years and without existing CVD, populated the model.

**Interventions** Four cardiovascular risk algorithms were assessed: (1) general population lipid, (2) general population body mass index (BMI), (3) SMI-specific lipid and (4) SMI-specific BMI, compared against no algorithm. At baseline, each cardiovascular risk algorithm was applied and those considered high risk (≥10%) were assumed to be prescribed statin therapy while others received usual care.

**Primary and secondary outcome measures** Quality-adjusted life years (QALYs) and costs were accrued for each algorithm including no algorithm, and cost-effectiveness was calculated using the net monetary benefit (NMB) approach. Deterministic and probabilistic sensitivity analyses were performed to test assumptions made and uncertainty around parameter estimates.

**Results** The SMI-specific BMI algorithm had the highest NMB resulting in 15 additional QALYs and a cost saving of approximately £53 000 per 1000 patients with SMI over 10 years, followed by the general population lipid algorithm (13 additional QALYs and a cost saving of £46 000).

**Conclusions** The general population lipid and SMI-specific BMI algorithms performed equally well. The ease and acceptability of use of an SMI-specific BMI algorithm (blood tests not required) makes it an attractive algorithm to implement in clinical settings.

For numbered affiliations see end of article.

**Correspondence to**
Dr David Osborn;
d.osborn@ucl.ac.uk

## Strengths and limitations of this study

► Health economic modelling employs mathematical modelling to extrapolate outcomes including both effects and costs beyond trial data, allowing the long-term effectiveness and cost-effectiveness of an intervention to be determined.

► To our knowledge, this is the first economic analysis to determine the effectiveness and cost-effectiveness of using cardiovascular disease (CVD) risk algorithms in patients with severe mental illness in primary care and subsequent treatment with statin therapy.

► A patient-level simulation model using real patient primary care data was developed, allowing the accumulating history of each individual to predict their transitions, costs and health outcomes.

► Deterministic and probabilistic sensitivity analyses were undertaken to account for variability in data inputs.

► The most widely used CVD risk algorithm in general practice in England was not used due to lack of availability of coefficients for the algorithm.

## BACKGROUND

People with severe mental illness (SMI), defined as schizophrenia, bipolar disorder and other non-organic psychotic conditions, have an increased risk of cardiovascular disease (CVD) compared with the general population.[1 2] This is due to an increased prevalence of modifiable cardiovascular risk factors[3–5], that some antipsychotic drugs may cause weight gain and abnormalities of lipid and glucose metabolism,[6 7] and that cardiovascular risk factors may be under-treated in people with SMI.[2] Up to 88% of people with schizophrenia have untreated dyslipidaemia and up to 66% have untreated hypertension.[3]

Cardiovascular risk algorithms are widely used in clinical practice to guide primary prevention CVD strategies,[8] in particular the initiation of lipid-modifying medication (statins). Guidelines for the initiation of statin therapy in clinical practice is country specific, with 10-year CVD risk thresholds of

7.5% being recommended in the USA,[9] 10% in Europe[10] and 10% recently recommended by the National Institute for Health and Care Excellence (NICE) in the UK.[11] A number of CVD risk algorithms exist,[12–14] including QRISK2, which is endorsed by NICE for use in primary care in the UK. The current Quality Outcomes Framework (QOF) incentivises annual monitoring and assessment of CVD risk using QRISK2 in primary care.[15] It also incentivises annual monitoring of blood pressure, alcohol and smoking status for patients with SMI.[15] Previously, QOF indicators also included measurements of total to high-density lipoprotein (HDL)-cholesterol ratio, glucose levels and weight in those with SMI.[16] Despite the presence of QOF indicators, monitoring of CVD risk remains low.[17 18] In addition, QRISK2 may incorrectly estimate CVD risk in some high-risk populations.[19] As a result, population-specific algorithms have been developed.[20–22] In 2015, an SMI-specific CVD risk algorithm, PRIM-ROSE, was developed and validated[23] using data from primary care attendees in the UK. In addition to variables common to other CVD risk algorithms, it includes psychiatric diagnosis, antipsychotic medication, harmful use of alcohol, use of antidepressants and social deprivation. The PRIMROSE algorithm is available as both a lipid model (including measures of total cholesterol and HDL cholesterol) and a body mass index (BMI) model (where lipid measures are replaced with measures of BMI). Both PRIMROSE models have been shown to perform better than the general population-based Framingham algorithm at predicting new CVD events.[23] However, the effectiveness and cost-effectiveness of these models in clinical practice are unknown.

The aim of this study is to evaluate the 10-year costs and consequences of an SMI-specific risk algorithm (PRIM-ROSE) compared with using general population CVD risk algorithms in the risk management and primary prevention of CVD in those with SMI. Our analysis was from a UK primary care population perspective using English healthcare costs.

## METHODOLOGY
### Study design
We developed a patient-level simulation to hypothetically model the progress of people with SMI over 10 years, accumulating the history of each individual to predict their transitions, costs and health outcomes. Real primary care data were used to capture the heterogeneity of the primary care SMI population. The model was created in Microsoft Excel 2010 in line with methodological recommendations for evaluations of new healthcare technologies and interventions.[24 25]

### Population
A cohort of 38 824 people with SMI was identified in The Health Improvement Network (THIN),[26] an anonymised longitudinal primary care database. THIN includes electronic medical records for more than 11 million individuals, registered with over 500 general practices in the UK. Available patient information includes demographics, local area deprivation (Townsend quintile), diagnoses, prescriptions, referrals, hospitalisations, laboratory tests, immunisations and clinical measures (eg, blood pressure, cholesterol). The demographics, prevalence of major conditions and mortality rates in THIN are similar to the UK general population.[27] Rates and demographics of people with SMI are also comparable to epidemiological estimates seen in previous studies of SMI.[28] Due to missing data, multiple imputation was used to generate 10 imputed data sets,[29] which were used to calculate transition probabilities of primary CVD events and all-cause mortality (see Transition probabilities section). Individuals in the extracted cohort had a recorded diagnosis of schizophrenia or schizoaffective disorder, bipolar disorder, other long-term psychotic illness (non-organic psychoses) and/or were listed on the SMI register[28]; were within the age limits of CVD risk assessment tools (30 to 74 years); and were free of CVD at their last point of contact (n=33 206),[23] where CVD was defined as a recorded diagnosis of coronary heart disease (CHD) including myocardial infarction (MI), angina, and major coronary surgery and revascularisation, or cerebrovascular disease (CVA) including haemorrhagic stroke, ischaemic stroke and transient ischaemic attack (TIA).

Due to computational complexities of our economic model, the patient-level simulation used a sample of 1000 patients randomly selected from one of the imputed datasets using the random number generator in Microsoft Excel. Data from patients' last known appointments (complete and imputed) were used as their baseline data in the model.

### CVD risk assessment tools
We calculated a CVD risk score for each of the 1000 patients using four different CVD risk algorithms in four separate analyses. The risk algorithms assessed are:
1. a general population lipid algorithm
2. a general population BMI algorithm
3. an SMI-specific lipid algorithm
4. an SMI-specific BMI algorithm.

Algorithms 1 and 2 are based on an adaptation of the widely used Cox Framingham algorithm,[14] herein referred to as the general algorithm, which was created and validated using THIN data. Algorithms 3 and 4 are derived from UK SMI patients in THIN, aged 30 to 90 years (PRIMROSE)[23] (online supplementary table 1). Performance of these algorithms has been previously tested.[23] Results demonstrated both SMI-specific algorithms (PRIMROSE) had higher, and therefore better, discrimination and calibration statistics than the general population algorithms. Calibration plots indicated both SMI-specific algorithms predicted CVD risk more accurately than the general population algorithms.

A fifth analysis using no CVD risk algorithm was included to estimate the costs and consequences of no intervention.

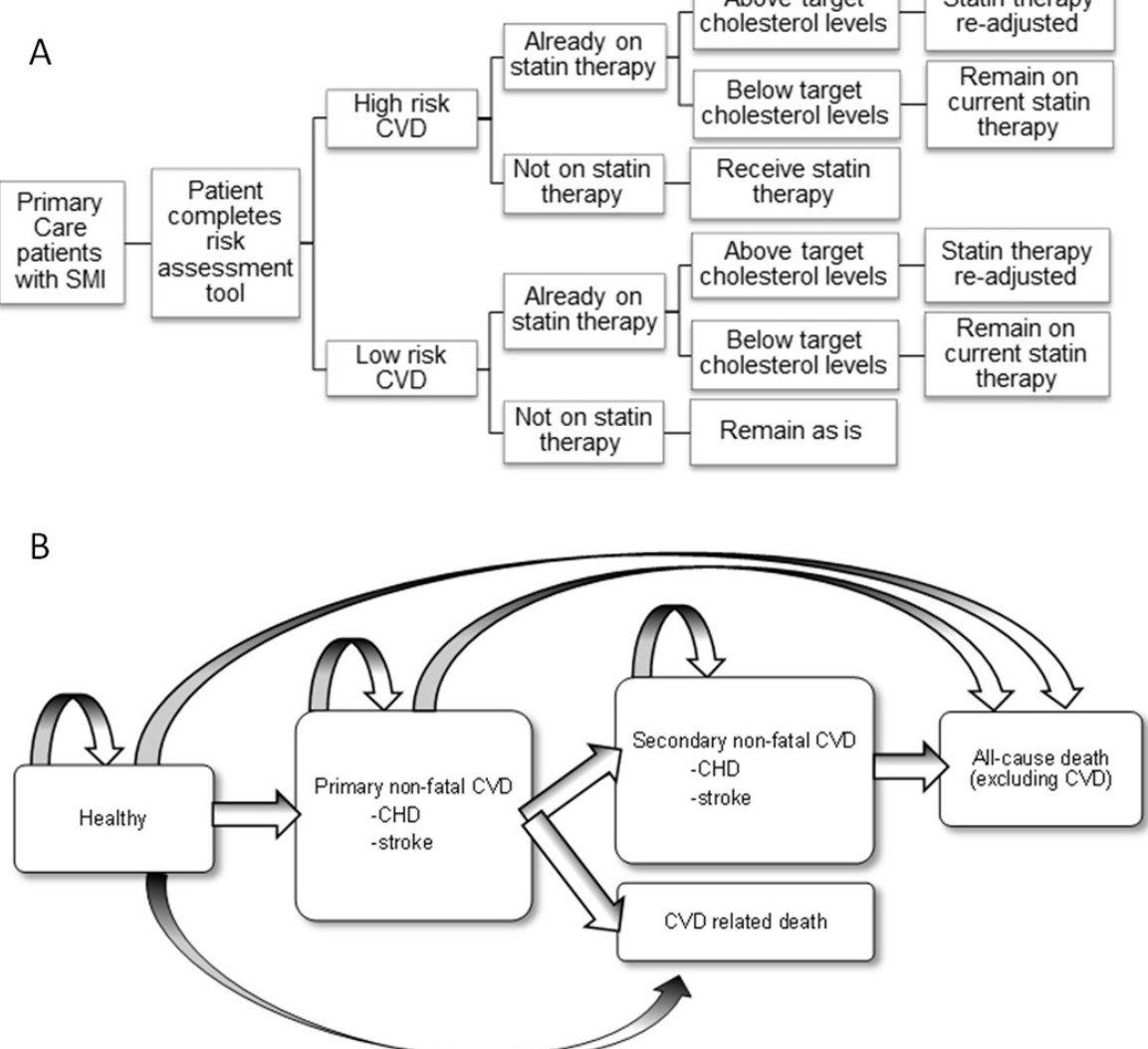

**Figure 1** (A) Decision tree of how those with severe mental illness (SMI) and at risk of cardiovascular disease (CVD) will be targeted for CVD risk management. (B) Decision tree of the possible health states and transitions in the economic model where non-fatal coronary heart disease (CHD) comprises stable angina, unstable angina, myocardial infarction, surgery and unclassified CHD; non-fatal stroke comprises transient ischaemic attack, haemorrhagic stroke, ischaemic/unclassified stroke and unspecified cerebrovascular disease.

## Model structure

The patient-level economic model includes (1) a decision tree to identify those at risk of CVD over 10 years and eligible for statin therapy (figure 1A), and (2) a Markov state transition model of 10 one-year cycles where patients can remain healthy, have a primary CVD event, have a secondary CVD event or die (figure 1B and Transition probabilities section).

At baseline, all patients in the economic model enter the decision tree and one of the four CVD risk algorithms described above is applied to calculate their 10-year CVD risk score. Those scoring over the CVD risk threshold (eg, 10%) are considered at high risk and receive statin therapy. Individuals classified as low risk are assumed not to receive statin therapy and remain in 'usual care'. Patients already on statin therapy before baseline remain on statins, regardless of their risk characteristics. However,

if a patient is already on statin therapy and their cholesterol levels for their last general practitioner (GP) consultation are above target levels (defined as total cholesterol >5 mmol/L or total cholesterol to HDL-cholesterol ratio >4 mmol/L),[30][31] their statin therapy is altered according to the average weighted changes that occur in statin prescribing as calculated from the THIN database. We applied the CVD threshold of 10% recently recommended by NICE to each of the models to reflect future clinical practice.[11] The 20% threshold as per current Quality and Outcomes Framework (QOF) guidelines[15] was also applied (see online supplementary appendix). In the no algorithm scenario, patients do not enter the decision tree.

After patients are classified as high risk and receive statin therapy, or low risk and receive usual care, all patients enter the economic model in a 'healthy', free of

CVD, state and continue to cycle through the economic model for 10 one-year cycles or until they die. Patient-level survival models[32] are used for each annual cycle to determine if a patient (1) remains in their current health state, (2) has a primary non-fatal CVD event, (3) has a secondary non-fatal CVD event, (4) has a fatal CVD event or (5) dies from non-CVD-related causes. All events occur at the beginning of a cycle.

This was repeated for each of the CVD risk algorithms in separate analyses.

### Transition probabilities

In the model, events were assumed to have occurred when the patient-specific probability of an event was more than a random number generated in Excel. For example, if a patient's probability of a primary CVD event in a cycle was 16.3% and the random number (taking possible values between 0 and 1) generated was 0.155 (15.5%), the patient was assumed to have had a primary CVD event. If, however, the random number generated was 0.45 (45.0%), the event did not occur. Events for patients were carried over cycles so that only patients who had a primary CVD event could have a secondary CVD event.

The probability of having a primary CVD event each year after baseline was calculated from the 10 imputed datasets of the SMI cohort using a survival model (Weibull distribution to allow calculation of time-dependent hazard functions).[32] Separate models were estimated for CHD and CVA. Development of these models was based on 38 824 people in THIN with SMI and aged over 18 years, with a maximum follow-up of 16 years. The covariates used included age, sex, systolic blood pressure, use of antihypertensive therapy, HDL cholesterol, use of cholesterol-lowering/cholesterol-altering therapy, height, weight, presence of diabetes, smoking status, history of heavy drinking, type of SMI, use of first-generation antipsychotic therapy, use of second-generation antipsychotic therapy, and history of depression or use of antidepressant therapy. The coefficient for each covariate was calculated from the 10 imputed datasets for each model and is reported in online supplementary table 2.

Primary CVD included both non-fatal and fatal events. A non-fatal CVD event comprised CHD and its substates, and CVA and its substates. The substates of a primary CHD event included stable angina, unstable angina, MI, coronary artery surgery and unclassified CHD, while a primary CVA event included TIA, haemorrhagic stroke, ischaemic/unclassified stroke and unspecified cerebrovascular disease. Non-fatal primary events were separated into substates of CHD and CVA so that costs and consequences could be allocated to each substate. The proportion of patients in each non-fatal CVD substate and the proportion of patients dying from a primary CVD event were equal to the proportion of patients in each group of these diagnostic categories found in the original cohort of people with SMI used to develop and test the risk algorithms in THIN (online supplementary table 3).

The probability of having a secondary CVD event was calculated from the model in the Reduction of Atherothrombosis for Continued Health (REACH) Registry.[33] The REACH algorithm is not SMI specific; however, it is an international equation to predict recurrent CVD based on patient-level characteristics over 20 months from a primary CVD event. Two equations are reported: one for secondary CVD and one for secondary fatal CVD. For each of the equations reported, the adjustment for country variable was omitted given that UK in the REACH algorithm is the comparator value and hence a coefficient for the UK is not reported. There was no information in our dataset regarding the number of vascular beds, presence of coronary heart failure (CHF), presence of atrial fibrillation (AF) and cardiovascular treatment with aspirin. For all patients in the model, it was assumed that there was only one vascular bed affected, CHF and AF were absent and they were not receiving cardiovascular treatment with aspirin. The 20-month estimate was converted to a 12-month estimate using the formula $p = 1 - \exp\{-rt\}$ where p is the probability, r is the rate and t is the time period of interest.[32]

Secondary CVD events comprised non-fatal and fatal events. Non-fatal secondary events were separated into substates of MI and stroke for cost and consequences purposes. The proportion of patients in each substate was based on percentages of people in each group in the REACH registry's 4-year follow-up data[34] (online supplementary table 3). As the proportion of patients in each substate for fatal secondary events was unknown, we assumed an equal proportional distribution.

The probability of dying from causes other than CVD was calculated using a survival model (Weibull distribution) and the THIN SMI population described in the Population section (online supplementary table 2).

### Effectiveness of CVD risk management strategy

The benefits of statin therapy were modelled by applying the relative risk reduction of CVD from statin use from a Cochrane review (0.73 and 0.78 for CHD and stroke, respectively)[35] to the predicted risks of CVD for all patients newly prescribed statins. As prescription of statins at baseline is a variable in the primary CVD event survival models estimated from THIN (online supplementary table 2), the risk of a primary CVD event for patients with a statin prescription at baseline was already modified.

### Costs

Costs included in our model were the cost of the CVD risk algorithm, CVD risk management and CVD events (online supplementary table 4). The cost of the risk algorithm comprised the cost of the time taken for the GP to complete the CVD risk prediction algorithm, which was estimated to be an additional 5 min on top of a regular consultation,[36] as well as the cost of a blood test[37] for the lipid algorithms. This was calculated as £20 for CVD risk algorithms requiring a blood test and £19 without. The cost of CVD risk management for those identified

as high risk comprised the cost of statin therapy (£21 per person per year). We assumed 20 mg of atorvastatin was prescribed, as per current QOF CVD prevention guidelines.[15] This was applied for the duration of the model, assuming 100% adherence with statin therapy. For patients already on statin therapy but with high cholesterol, a weighted average change in prescription costs was applied. All prescription costs were taken from the British National Formulary.[38]

The costs of fatal and non-fatal CVD events were extracted from an economic evaluation of statins for primary prevention of coronary events.[39] The cost of CHD surgery was calculated from the weighted mean of reference costs for CHD surgical operations.[37] The cost of unclassified CHD and unspecified CVA was calculated as the average of all CHD and CVA events, respectively, given there is no information on the cost of unclassified and unspecified events. All costs were inflated to 2012/2013 values using conversion rates in Curtis.[36]

## Outcomes

The mortality and morbidity impact was evaluated using quality-adjusted life years (QALYs) as recommended by NICE in the UK.[40] QALYs are calculated by multiplying a utility score (preference based value of a health state of an individual) by the amount of time in that health state. A utility score of 1 represents perfect health and 0 death. All patients in the model were assumed to have the utility score of someone whose SMI symptoms are being managed (0.865).[41] If a patient had a non-fatal CVD event, a utility decrement was applied (online supplementary table 4). This was applied for the year of the event and every year thereafter, until the end of the model or the patient died. The utility decrements associated with non-fatal CVD events of angina, MI, TIA and stroke were taken from the same economic model as CVD costs mentioned in the Costs section. Where utility decrements were unknown, we assumed the utility decrement associated with CHD surgery was the same as MI; the utility decrement associated with unclassified CHD was the weighted average of stable angina, unstable angina and MI; and the utility decrement associated with unspecified CVD was the weighted average of stroke and TIA.

Cost-effectiveness was calculated using the net monetary benefit (NMB) approach.[32] The NMB is defined as the total discounted QALYs for 1000 patients over 10 years, multiplied by a given willingness to pay, minus the total discounted cost for 1000 patients over 10 years, where the willingness to pay is the maximum monetary value a decision maker is willing to pay for a QALY. The scenario with the highest NMB is the preferred option. We tested willingness to pay values of £20 000 and £30 000 per QALY from the results of the probabilistic sensitivity analysis.[40] Cost-effectiveness acceptability curves were constructed to calculate the probability that each algorithm had the highest NMB for a range of values of willingness to pay for a QALY.

All future benefits (QALYs) and costs were discounted at 3.5% per annum.[40]

## Sensitivity analyses

Deterministic and probabilistic sensitivity analyses were performed to test assumptions made and uncertainty around parameter estimates. Variables in the probabilistic sensitivity analyses, CIs and distributions are reported in online supplementary table 4. For cost inputs, where CIs were not reported, we assumed the SD was equal to the mean, as recommended by Briggs *et al*.[32]

One-way sensitivity analyses included a base case deterministic analysis where all input parameters with variability were held at their mean value and subsequent analyses varying a single input to test the assumptions made (while all other input parameters remained at their mean value). We tested the following assumptions using 5000 iterations for each analysis:

► All costs, treatment costs associated with CVD risk management with statin therapy, intervention costs of using the CVD risk algorithms and cardiovascular event costs were doubled in separate analyses to explore the potential underestimation of costs in our model.
► The utility associated with SMI was reduced to represent relapse (0.479) and SMI with extrapyramidal symptoms, a drug-induced movement disorder with acute and tardive symptoms (0.604)[41] in separate analyses.
► The treatment effect of statin therapy was reduced to the upper OR of the 95% CI published in the Cochrane review of the effect of statin therapy on CVD in the general population[37] to explore potential differences that may be present with effectiveness of statin therapy in an SMI population compared with the general population. These values were 0.8 and 0.89 for CHD and stroke, respectively.
► Adherence with statin therapy was reduced to 50% in line with rates of non-adherence with statin therapy.[42]

The probabilistic sensitivity analysis was conducted in line with Decision Support Unit guidance[43] for patient-level simulations with 100 inner loops for the patient-level simulation and 1000 outer loops for the probabilistic sensitivity analysis. The model values for each of the 1000 outer loops were calculated from the mean of each inner loop.

## RESULTS
### Patient characteristics
Baseline characteristics of the 1000 patients and the total eligible cohort are reported in table 1.

### Classification of those at high risk
Table 2 and online supplementary table 5 summarise the proportion of patients classified as 'high risk' of CVD by the four algorithms at 10% and 20% thresholds, respectively. The SMI-specific BMI algorithm classified the

**Table 1** Baseline characteristics for extracted SMI population who were free of cardiovascular disease (CVD) and aged 30–74 years and sample of 1000 patients, where continuous variables are reported as mean (SD) and discrete variables are reported as n (%)

| Baseline characteristics | Total population | Sample of population |
|---|---|---|
| n | 33 026 | 1000 |
| Age, mean (SD), years | 50.3 (12.0) | 50.2 (12.0) |
| Female, n (%) | 16 155 (48.7) | 513 (51.3) |
| Type of SMI, n (%) | | |
| Schizophrenia | 11 495 (34.8) | 335 (33.5) |
| Bipolar disorder | 8822 (26.7) | 256 (25.6) |
| Other non-organic psychotic disorders | 9098 (27.6) | 313 (31.3) |
| On SMI registry but no diagnoses | 3611 (10.9) | 96 (9.6) |
| SBP, mean (SD), mm Hg | 128 (16) | 128 (16) |
| Antihypertensive therapy, n (%) | 5402 (16.4) | 164 (16.4) |
| Total cholesterol, mean (SD), mmol/L | 5.4 (1.1) | 5.3 (1.0) |
| HDL cholesterol, mean (SD), mmol/L | 1.3 (0.4) | 1.4 (0.4) |
| Lipid-lowering therapy, n (%) | 3545 (10.7) | 97 (9.7) |
| Weight, mean (SD), kg | 80.0 (18.9) | 79.5 (18.8) |
| Height, mean (SD), m | 1.7 (0.1) | 1.7 (0.1) |
| BMI, mean (SD), kg/m$^2$ | 28.0 (6.1) | 27.9 (6.0) |
| Diabetes, n (%) | 2412 (7.3) | 74 (7.4) |
| Smoking status, n (%) | | |
| Non-smoker | 11 474 (34.7) | 355 (35.5) |
| Ex-smoker | 3726 (11.3) | 100 (10.0) |
| Current smoker | 17 826 (54.0) | 545 (54.5) |
| History of heavy drinking, n (%) | 4706 (14.3) | 139 (13.9) |
| Depression, n (%) | 21 190 (64.2) | 633 (66.3) |
| Antidepressant therapy, n (%) | 13 055 (39.5) | 377 (37.7) |
| First-generation antipsychotic therapy, n (%) | 4982 (15.1) | 133 (13.3) |
| Second-generation antipsychotic therapy, n (%) | 10 691 (32.4) | 311 (31.1) |
| Townsend score of deprivation*, n (%) | | |
| 1 | 4886 (14.8) | 143 (14.3) |
| 2 | 5332 (16.2) | 158 (15.8) |
| 3 | 6639 (20.1) | 183 (18.3) |
| 4 | 8048 (24.4) | 269 (26.9) |
| 5 | 8121 (24.6) | 247 (24.7) |
| Calendar year†, mean (SD) | 2007.7 (3.5) | 2007.7 (3.5) |

*Townsend score of deprivation is an index made up of unemployment, overcrowding, non-car ownership and non-home ownership, where 1 represents lower degree of deprivation and 5 represents higher degree of deprivation.
†Calendar year refers to the calendar year in which the baseline data were collected to account for any time trends.
BMI, body mass index; HDL, high-density lipoprotein; SBP, systolic blood pressure; SMI, severe mental illness.

highest number of patients as 'high risk' of CVD (326 patients at 10% and 117 at 20%) and resulted in the greatest number of new statin prescriptions (255 patients at 10% and 81 at 20%). The general BMI algorithm classified the lowest number of patients as 'high risk' (222 patients at 10% and 65 at 20%) and generated the lowest number of new statin prescriptions (175 patients at 10% and 44 at 20%).

**Clinical and cost outcomes**

The number of CVD events, cost and QALYs per 1000 patients with SMI over 10 years for each algorithm (including no algorithm) is reported in table 3. At the 10% threshold in 1000 patients over 10 years, a CVD risk algorithm plus statin treatment prevents a minimum of 9 (general BMI algorithm) and maximum of 13 (SMI-specific BMI algorithm) primary CVD events (one fatal) and

**Table 2** Number of people (out of 1000) classified as high and low risk by the various CVD risk algorithms at a CVD risk threshold of 10%; further stratified by use of statin therapy at baseline

| | Algorithm | | | |
|---|---|---|---|---|
| | **General lipid algorithm** | **SMI-specific lipid algorithm** | **General BMI algorithm** | **SMI-specific BMI algorithm** |
| High risk (>10%) | | | | |
| Total | 268 | 241 | 222 | 326 |
| Currently prescribed statins | 58 | 59 | 47 | 71 |
| Not currently prescribed statins | 210 | 182 | 175 | 255 |
| Low risk (<10%) | | | | |
| Total | 732 | 759 | 778 | 674 |
| Currently prescribed statins | 39 | 38 | 50 | 26 |
| Not currently prescribed statins | 693 | 721 | 728 | 648 |

BMI, body mass index; CVD, cardiovascular disease; SMI, severe mental illness.

three to four secondary CVD events across all models. This is equivalent to a 4%–6% reduction in primary CVD events and a 12%–16% reduction in secondary CVD events. The 20% model prevents three to five primary events (0 to 1 fatal) and one to two secondary events (online supplementary table 6).

The number of events stratified by risk and statin therapy at baseline is reported in online supplementary table 7.

All four CVD risk algorithms result in more QALYs for less cost compared with when no algorithm and no additional statin therapy is given (table 3). The SMI-specific BMI algorithm has a higher NMB (£43 797 representing 0.03% of the total NMB) than the general lipid algorithm and all other algorithms. The SMI-specific BMI algorithm has the highest NMB for 45% of iterations of the probabilistic sensitivity analysis at a willingness-to-pay value of £20 000 (figure 2). The results for the 20% threshold are similar (online supplementary figure 1).

Results of subanalyses and deterministic analyses are reported in the online supplementary appendix (online supplementary Results 1.1–1.2 including online supplementary figures 2, 3 and table 8).

## DISCUSSION

This is the first study to model the long-term effectiveness and cost-effectiveness of a CVD risk algorithm plus risk management strategy in people with SMI. Prescribing statins to patients with SMI in primary care with a CVD risk score over 10% resulted in a 4%–6% reduction in primary CVD events and a 12%–16% reduction in secondary CVD events over 10 years. The provision of a relatively low-cost identification tool (the risk algorithm) and relatively low-cost intervention (statins) compared with the high cost of CVD events means that the intervention saves up to £53 000 per 1000 patients over 10 years or £53 per patient administered a CVD risk algorithm. Using a 10% threshold for identifying high-risk patients

resulted in fewer CVD events than the 20% threshold and hence greater cost savings.

The aim of our economic modelling strategy was to identify if there is any added value in using an SMI-specific risk algorithm, rather than standard general population risk scores, for CVD prevention in people with SMI. The best-performing risk assessment tool was the SMI-specific BMI algorithm. This may have been a result of its classification of more individuals at high risk of CVD and eligible for statin therapy than other algorithms. Differences between this algorithm and the general population lipid algorithm were minimal, with the SMI-specific BMI algorithm resulting in an additional two QALYs compared with the general lipid algorithm, at an additional cost saving of approximately £6000 per 1000 individuals over 10 years. Given there is little to no difference between the two tools economically, the decision regarding which algorithm to use in routine clinical practice becomes one of implementation, advocacy and ease of use. One could argue in favour of using a general population-derived lipid model as these are already used in UK general practice and hence require no change. On the other hand, the SMI-specific BMI model, although potentially requiring additional training and implementation costs, could confer additional benefit by raising awareness of the need to improve CVD outcomes in people with SMI and providing a model that requires no blood test to estimate risk, a limitation of other CVD risk algorithms as many people, with and without SMI, decline blood tests.[44] The ease of implementation and delivery of the SMI-specific BMI model means it could be used in any setting, including mental healthcare and non-clinical settings without blood results. This is particularly important as many people with SMI do not attend primary care and monitoring of CVD risk factors remains low in other settings.[45–49] The SMI-specific BMI model provides an opportunity to target more people with SMI, to increase identification of those at high risk of CVD and decrease the physical, social and financial burden associated with CVD.

**Table 3** Costs, QALYs, NMBs and number of events per 1000 individuals for each CVD algorithm (including no algorithm) when a CVD risk threshold of 10% was employed

| Outcomes | Algorithm | | | | |
| --- | --- | --- | --- | --- | --- |
| | General lipid algorithm | SMI-specific lipid algorithm | General BMI algorithm | SMI-specific BMI algorithm | No algorithm |
| Costs and QALYs, mean (95% CI) | | | | | |
| Costs of administering algorithm | 20 006 (19 906 to 20 106) | 20 006 (19 906 to 20 106) | 19 010 (18 935 to 19 085) | 19 010 (18 935 to 19 085) | 19 010 (18 935 to 19 085) |
| Costs of new statin prescriptions | 38 371 (37 849 to 38 892) | 33 465 (33 012 to 33 919) | 31 611 (31 183 to 32 040) | 47 152 (46 510 to 47 794) | n/a |
| Costs of CVD events | 1 871 508 (1 698 400 to 2 044 617) | 1 882 462 (1 708 721 to 2 056 202) | 1 891 266 (1 717 495 to 2 065 038) | 1 855 697 (1 683 643 to 2 027 751) | 1 985 044 (1 807 487 to 2 162 602) |
| Total costs undiscounted | 1 929 885 (1 756 824 to 2 102 946) | 1 935 933 (1 762 235 to 2 109 631) | 1 941 887 (1 768 154 to 2 115 621) | 1 921 859 (1 749 857 to 2 093 861) | 1 985 044 (1 807 487 to 2 162 602) |
| Total costs discounted | 1 666 228 (1 515 958 to 1 816 499) | 1 671 497 (1 520 650 to 1 822 345) | 1 676 569 (1 525 676 to 1 827 462) | 1 659 340 (1 509 988 to 1 808 692) | 1 712 136 (1 557 767 to 1 866 506) |
| QALYs discounted | 6828 (6 813 to 6 843) | 6827 (6 812 to 6 842) | 6826 (6 811 to 6 841) | 6830 (6 815 to 6 845) | 6815 (6 800 to 6 831) |
| Cost compared with no algorithm | –45 908 | –40 639 | –35 567 | –52 797 | |
| QALYs compared with no algorithm | 13 | 12 | 11 | 15 | |
| Net monetary benefit, mean (95% CI) | | | | | |
| £20 000 WTP threshold | 134 898 309 (134 467 161 to 135 329 457) | 134 872 660 (134 439 483 to 135 305 838) | 134 841 184 (134 407 261 to 135 275 106) | 134 942 106 (134 513 538 to 135 370 673) | 134 593 353 (134 147 224 to 135 039 482) |
| £30 000 WTP threshold | 203 180 577 (202 601 927 to 203 759 228) | 203 144 739 (202 563 347 to 203 682 603) | 203 100 060 (202 517 517 to 203 682 603) | 203 242 828 (202 667 604 to 203 818 053) | 202 746 098 (202 146 930 to 1 to 203 345 265) |
| Events, mean (95% CI) | | | | | |
| Primary non-fatal CHD | 81.87 (75.70 to 88.05) | 82.53 (76.33 to 88.74) | 82.98 (76.76 to 89.21) | 81.25 (75.14 to 87.37) | 87.57 (81.12 to 94.02) |
| Primary fatal CHD | 9.26 (8.56 to 9.95) | 9.33 (8.63 to 10.03) | 9.38 (8.68 to 10.09) | 9.18 (8.49 to 9.87) | 9.89 (9.16 to 10.62) |
| Primary non-fatal stroke | 104.18 (94.50 to 113.86) | 104.64 (94.93 to 114.35) | 104.92 (95.21 to 114.63) | 103.43 (93.80 to 113.06) | 108.77 (98.91 to 118.63) |
| Primary fatal stroke | 7.25 (6.56 to 7.93) | 7.27 (6.58 to 7.95) | 7.30 (6.62 to 7.99) | 7.18 (6.50 to 7.85) | 7.55 (6.86 to 8.24) |
| Secondary non-fatal CVD | 14.77 (13.73 to 15.81) | 14.97 (13.92 to 16.03) | 15.15 (14.09 to 16.21) | 14.48 (13.46 to 15.50) | 17.20 (16.03 to 18.38) |
| Secondary fatal CVD | 6.53 (6.10 to 6.96) | 6.64 (6.20 to 7.07) | 6.65 (6.22 to 7.09) | 6.41 (5.99 to 6.83) | 7.78 (7.28 to 8.27) |
| Death from other causes | 119.43 (118.63 to 120.24) | 119.40 (118.59 to 120.21) | 119.35 (118.55 to 120.16) | 119.40 (118.59 to 120.20) | 119.05 (118.25 to 119.85) |

Discounted costs and QALYs reflect time preference for current benefits over future ones.
BMI, body mass index; CHD, coronary heart disease; CVD, cardiovascular disease; NMB, net monetary benefit; QALYs, quality-adjusted life years; SMI, severe mental illness; WTP, willingness to pay.

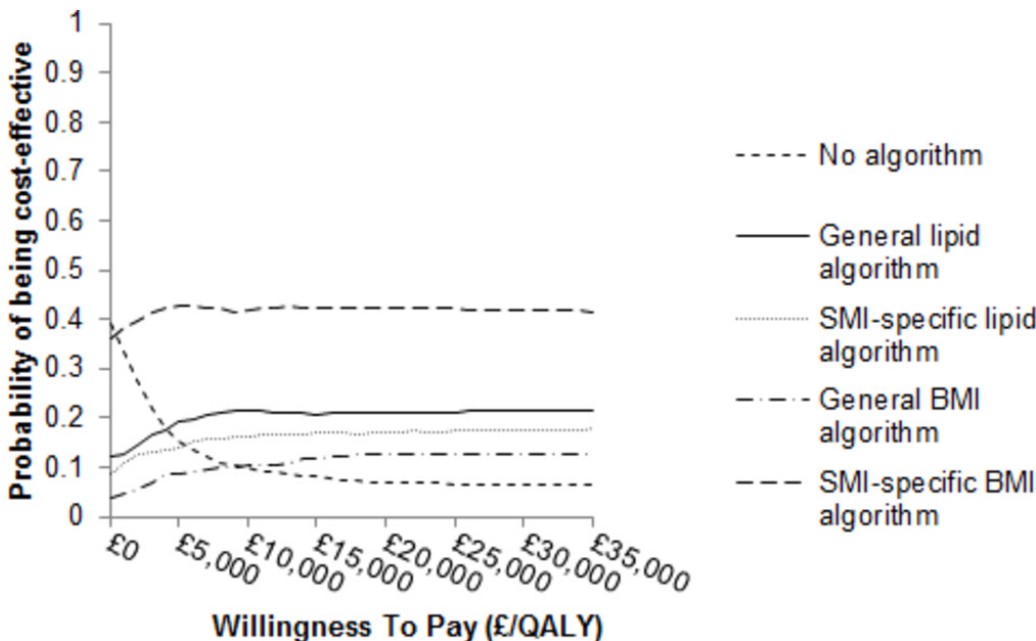

**Figure 2** Cost-effectiveness acceptability curves of each CVD risk algorithm, compared to no algorithm, when a CVD risk threshold of 10% was employed. BMI, body mass index; CVD, cardiovascular disease; QALY, quality-adjusted life year; SMI, severe mental illness.

While current CVD prevention guidelines are based on CVD risk assessment using risk algorithms, there are other ways to target CVD prevention including identification of CVD risk factors for CVD risk management. We are currently evaluating methods of identifying CVD risk factors in those with SMI in primary care settings in England and decreasing CVD risk via a nurse-led care intervention.[50]

## STRENGTHS AND LIMITATIONS

CVD events in primary prevention populations are rare requiring large sample sizes and long follow-up periods to be able to show statistically significant differences in CVD events between trial arms. Therefore, most primary prevention CVD trials use proxy outcomes, such as lipid levels, to determine effectiveness rather than the prevention of actual CVD events. This can limit the conclusions that can be drawn for economic evaluations of primary CVD prevention interventions using trial data only, given the cost, morbidity and mortality implications of CVD events. Our economic model/analysis has the strength of using real patient-level primary care data for patients to model the long-term costs and consequences of a CVD primary prevention intervention, making it a better representation of real life. Few economic evaluations using patient-level simulations have attempted to use primary care data before, and none have been performed within SMI populations despite their high cardiovascular risk.

There were some weaknesses in our model/analysis. First, we assumed that adherence with statin therapy is the same as that seen in clinical trials. Overall, statins tend to have high rates of non-adherence[42] and adherence is likely to be higher in trials[51] where a gold standard of

clinical care is provided, participants are monitored more closely and those recruited are generally predisposed to follow advice about medication. When we tested this in a deterministic analysis, assuming an adherence of 50%, the SMI-specific BMI and general lipid algorithms had the highest NMB at £20 000 per QALY. The effects of statin therapy were also taken from a systematic review and meta-analysis in a general population. This was tested in our deterministic analyses assuming a lower level of effectiveness. We also assumed the benefits of statin therapy were constant over 10 years, which may not be true.

Second, we were unable to obtain the coefficients for the algorithm used most widely in general practice in England, QRISK2.[12] Instead, we used the Framingham CVD risk prediction algorithm, but re-estimated to the UK general population. We cannot be sure this version is equivalent to QRISK2 in predicting risk, although previous analyses showed the re-estimated Framingham performed well in the SMI population.[23] Further to this, previous studies have reported that it is unlikely that using a different general population CVD risk algorithm will have a significant impact on the results of a cost-effectiveness analysis.[52]

It was assumed that all people with SMI were in a 'stable' mental state free of symptoms, which might not be realistic. This was done for simplicity and because the focus was reduced CVD events and not improved SMI treatment. Our deterministic models demonstrated that reducing the assumed utilities for patients with SMI did not have a significant impact on the results.

Third, we were unable to validate the economic model externally as there was no other data source on CVD events in primary care patients with SMI. An internal validation

comparing the number of CVD events recorded in the THIN data and the number of CVD events predicted when no algorithm was employed showed comparable results. Therefore, while our model population was considerably smaller than the larger cohort of 38 824 individuals, it was representative of the SMI population in THIN.

## CONCLUSIONS

This is the first economic model/analysis to quantify the costs and consequences of assessing patients with SMI in primary care with a CVD risk algorithm and prescribing statins to those classified as high risk. Our model suggests that there is a significant economic benefit associated with the improved management of modifiable CVD risk factors for patients with SMI, using statins. The SMI-specific BMI algorithm functioned better than the other CVD risk algorithms tested. The ease and acceptability of use for patients (due to lack of blood test) and potential to increase awareness of CVD risk in patients with SMI make it an attractive algorithm to implement in a range of settings. Once implemented, re-evaluation and comparison of the SMI-specific BMI algorithm to current practice using real-life data is necessary, as this has the potential to influence continuity of care for people with SMI at risk of CVD in the UK.

**Author affiliations**
[1]Department of Epidemiology and Preventive Medicine, Faculty of Medicine, Nursing and Health Sciences, Monash University, Melbourne, Australia
[2]Department of Primary Care and Population Health, Faculty of Population Health Sciences, University College London, London, UK
[3]Division of Psychiatry, Faculty of Brain Sciences, University College London, London, UK
[4]Camden and Islington National Health Service Foundation Trust, London, UK
[5]Human Development and Health Academic Unit, Faculty of Medicine, University of Southampton, Southampton, UK
[6]Department of Applied Health Research, Faculty of Population Health Sciences, University College London, London, UK
[7]Department of Statistical Science, Faculty of Mathematical and Physical Sciences, University College London, London, UK

**Contributors** EZ and RMH had full access to all the data in the study and take responsibility for the integrity of the data and the accuracy of the data analysis. Study concept and design: DO, SM, RMH. Acquisition, analysis or interpretation of data: EZ, RMH, RB, SH, KW, LM. Drafting of the manuscript: EZ, RMH, DO, IN. Critical revision of the manuscript for important intellectual content: all authors. Statistical analysis: EZ, RMH. Obtained funding: DO IN KW RIGH MK IP. Administrative, technical or material support: RB, AB, SH, RIGH, MK, LM, RO, IP, KW. Study supervision: RMH, DO.

**Funding** This paper summarises independent research supported by the National Institute for Health Research (NIHR) under its Programme Grants for Applied Research Programme (grant reference number: RP-PG-0609-10156). The funder had no role in the design and conduct of the study; collection, management, analysis and interpretation of the data; preparation, review or approval of the manuscript; and decision to submit the manuscript for publication. Prof Osborn is supported by the UCLH NIHR BiomedicalResearch Centre and he was also in partsupported by the National Institute for Health Research (NIHR) Collaborationfor Leadership in Applied Health Research and Care (CLAHRC) North Thames atBart's Health NHS Trust

**Disclaimer** The views expressed are those of the authors and not necessarily those of the National Health Service, the NIHR or the Department of Health.

**Competing interests** None declared.

**Provenance and peer review** Not commissioned; externally peer reviewed.

**Data sharing statement** No additional data available.

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
