## [Reviewer comments · BMJ Open]

ARTICLE DETAILS

TITLE (PROVISIONAL)	Effectiveness and cost-effectiveness of a cardiovascular risk prediction algorithm for people with severe mental illness (PRIMROSE).
AUTHORS	Zomer, Ella; Osborn, David; Nazareth, Irwin; Blackburn, Ruth; Burton, Alexandra; Hardoon, Sarah; Holt, Richard; King, Michael; Marston, Louise; Morris, Stephen; Omar, Rumana; Petersen, Irene; Walters, Kate; Hunter, Rachael

VERSION 1 - REVIEW

REVIEWER	Seren Roberts Bangor University Wales, UK.
REVIEW RETURNED	28-Jul-2016

GENERAL COMMENTS	This is an important topic area for research and will be of interest to a wide audience. Improving the way in which we assess and manage CVD risk for individuals with SMI in a more targeted way is of great value to the health community given the significant health inequalities experienced by these individuals. This study comes part way to this end. The study provides a well-considered computation model of CVD risk by using real patient data to generate simulated CVD trajectories for an SMI sample population. These simulated patient data sets were used to compare the strength of four different CVD risk algorithms (2 general population based and 2 SMI specific) in assessing CVD risks and resulting in improved management. A cost analysis was also carried out to determine the most cost effective algorithm for use in practice. Two algorithms (one general population and one SMI-specific) were found to be superior in terms of assessing and managing risk and were deemed cost effective strategies. These findings are likely to inform practice by enabling clinicians to consider more SMI-specific strategies for assessing and managing CVD risk in this disadvantaged group. Whilst the study is a computational model, the study does add to the body of knowledge by demonstrating the value of SMI-specific assessments of CVD risk and consequent improved management. Even though the general population lipid algorithm is shown to be as effective as the SMI-specific algorithm at identifying CVD risk, the latter does not require a blood test making it a valuable and practical approach to implement in practice. Overall the article is well written and concise. The article is original in that it provides some evidence that there may be an advantage to utilising SMI-specific CVD risk algorithms for this high risk group over general population algorithms (given the same outcome with
---

less burden on the patient). In addition, the simulations are based on real patient data sets and existing population parameters, thus strengthening the robustness and real-world applicability of the model.

The research question is clearly defined and answered and the design is appropriate and robust. There is a clear justification for the topic and the utility of algorithms for the assessment and subsequent management of CVD risk. However, a brief justification for use of computational modelling approach is needed in the background (first paragraph of the study strength and limitations perhaps). The aim refers to English NHS and I am uncertain of the justification for this above UK-wide NHS. A UK population dataset was used for the model and there is no indication the cost calculations related to NHS England only. This needs to be justified. Similarly, there is no reference to the source tariffs for the costs of GP time to complete the algorithm.

The population data set is well described and appropriate. The sample used to generate simulations are representative. However, a comment to reassure the audience that ethical approval was in place for the use of the THIN dataset is needed.

In addition, since there are three main elements that feed into the model (simulated patient data sets, comparison of 4 different algorithms, and the cost data) the organisation of the information in Section 2 methodology could be improved to make clear these main elements. An overall schematic of the model/process flow diagram would also help make this complex topic more accessible. Two useful figures/images have been included (decision tree and health states/transitions) which I have been therefore unable to find reference to in the text.

The results are well described with sufficient tables and eTables to support the report. Statistical significance is not reported for any result to demonstrate the strength of the findings in relation to the four algorithms or economic evaluations.

The interpretation and conclusion of the findings are appropriate. It would be useful for the authors to comment further about and discuss possible explanations for why the SMI-lipid algorithm and general population-BMI algorithm did not perform as well as the general population-lipid algorithm and the SMI-specific BMI algorithm. Minor typographical errors which would be eliminated though thorough proof reading.

The abstract summarises the study well although the focus here is on the economic element despite the title referring to effectiveness and cost-effectiveness of the algorithms. The authors refer to the model as an economical model but it seems to me to be more given work compares the effectiveness and cost-effectiveness of the algorithms.

Overall a very interesting and important paper.

REVIEWER	Rohini Mathur LSHTM, UK
REVIEW RETURNED	01-Aug-2016

GENERAL COMMENTS	This paper details study examining the effectiveness and cost effectiveness of four different CVD risk score algorithms, 2 designed for the general population, and 2 specific to individuals with serious mental illness (SMI), in comparison to no algorithm. Outcomes include QALYs, costs per algorithm, and cost-effectiveness using a net monetary benefit approach. The study showed important differences between each of the algorithms, with the general population lipid and SMI population BMI algorithm performing the best. Overall this is a well conducted study which shows important findings that general population algorithms can work well in a high CVD risk population, but that an SMI population specific algorithm (BMI) could be used alternatively, with the benefits that it is blood-free. Specific comments: Introduction: Line 26: The sentence specifies that most algorithms use a cut off of 7.5% or 10%, however to clarify the authors should expand to say that 7.5% risk of developing CVD within 10 years etc... The aim should make clear that the study is looking at both primary and secondary CVD. Some discussion on whether the algorithms chosen are appropriate for use with secondary CVD would be useful. Since the study is set in UK primary care, some description of the primary care context is warranted. For example, which CVD risk scores are recommended for use in primary care, whether these are used in SMI populations. I.e. Are people with SMI eligible for the annual health check program which is supposed to capture QRISK once a year. Methods: Line 51-54: It would be useful for the authors to discuss any existing literature around whether PRIMROSE has been found to perform better or worse in SMI populations than other risk scores (if this paper is the first to examine this, then this theme should be brought out earlier in the paper) The methodology needs to state earlier on which conditions are used to define CVD- I know this appears later in the description of the simulation models on page 11, but this is worth stating in the opening where the other variables are described. The methodology should also reference/signpost how SMI and CVD were defined (QOF definition, other?)- which code lists, algorithms were used and how can these be found in order to replicate the disease definitions in other settings (either from other published studies, or to an appendix/online resource). The authors state they identified fatal CVD- was this identified from the primary care record or from linked HES/ONS? If so, was linkage
--

	complete for all included patients? Discussion: This is an important study which highlights the utility of a blood-free algorithm in primary care settings. The study used 1000 patients due to computational limitations, it would be useful to discuss how the use of more patients/complete patient population would have changed the results. What biases may have been potentially introduced by using a smaller sample. Some discussion on how implementation of the SMI-BMI algorithm might be incorporated into current primary care practice/guidelines would give this paper important context. For example, what would the benefits be (if any) over and above existing practices, given that CVD risk assessment is already a part of standard clinical practice and incentivised by QOF. What factors would help practitioners decide whether to invest the time/training in using the SMI specific BMI algorithm. The authors stated that they couldn't compare against the most widely used risk scores due to lack of variable availability. Would it have been possible to explore these algorithms using multiple imputation for the missing variables- in order to give a comparison to risks scores already being used in current practice? This would be of great interest to clinical practitioners, for whom this study should help in their decision to move from current practice to potentially adopting a new SMI population specific score. Some wider discussion on the challenges of CVD Risk management in SMI populations would strengthen the paper. For example, differences in attendance, adherence, opportunities to identify CVD risk factors may be more important in managing CVD risk than a tailored risk algorithm. Some discussion on further work would be useful, would an observational cohort study/trial exploring the PRIMROSE risk scores in practice be feasible/useful?
--	---

VERSION 1 – AUTHOR RESPONSE

Reviewer 1 stated:

1. A brief justification for use of computational modelling approach is needed in the background (first paragraph of the study strength and limitations perhaps).

This has been amended and added into the strengths and limitations section of our manuscript.

2. The aim refers to English NHS and I am uncertain of the justification for this above UK-wide NHS. A UK population dataset was used for the model and there is no indication the cost calculations related to NHS England only. This needs to be justified.

We thank you for this oversight. Our analysis was from a UK primary care population perspective using English health care costs. This has now been changed.

3. No reference to the source tariffs for the costs of GP time to complete the algorithm.

The cost of GP time to complete the algorithm was sourced from Curtis L. Unit costs of health and social care 2013. United Kingdom: Personal Social Services Research Unit, 2013 (see reference 32).

It was estimated that completion of the algorithm would take approximately 5 minutes in addition to a regular consultation. The time specifics have now been added to the section 2.7 Costs of the manuscript.

4. A comment to reassure the audience that ethical approval was in place for the use of the THIN dataset is needed.

As the THIN dataset is a de-identified dataset, ethics approval to use this dataset was not needed. We, therefore, did not feel it was necessary to include a statement regarding ethical approval. However, previous work from our group, which included use of the THIN dataset to develop the PRIMROSE risk scores was approved by the CMD Medical Research's Scientific Review Committee (Osborn DP, Hardoon S, Omar RZ, et al. Cardiovascular risk prediction models for people with severe mental illness: Results from the Prediction and Management of Cardiovascular Risk in People with Severe Mental Illnesses (PRIMROSE) Research Program. *JAMA Psychiatry* 2015; 72: 143-151).

5. Since there are three main elements that feed into the model (simulated patient data sets, comparison of 4 different algorithms, and the cost data) the organisation of the information in Section 2 methodology could be improved to make clear these main elements. An overall schematic of the model/process flow diagram would also help make this complex topic more accessible. Two useful figures/images have been included (decision tree and health states/transitions) which I have been therefore unable to find reference to in the text.

Reference to the figures depicting the decision and Markov state transition model is included in text in section 2.4 Model Structure of the manuscript.

6. Statistical significance is not reported for any result to demonstrate the strength of the findings in relation to the four algorithms or economic evaluations.

Effectiveness and cost-effectiveness outcomes from economic models do not provide information on statistical significance. Statistical significance is also difficult to report in incremental cost-effectiveness ratios because of the combination of cost and health outcome. In line with best practice (Briggs AH & Gray AM. Handling uncertainty in economic evaluations of healthcare interventions. *BMJ* 1999; 319: 635-638) we have reported the probability that each model is the best which takes into account the uncertainty associated with all of the parameters in the model.

7. Useful for the authors to comment further about and discuss possible explanations for why the SMI-lipid algorithm and general population-BMI algorithm did not perform as well as the general population-lipid algorithm and the SMI-specific BMI algorithm.

The performance of the algorithms relies heavily on classification of people correctly who are high and low risk. The SMI-lipid algorithm and general population-BMI algorithm did not classify as many individuals at high risk as the SMI-BMI algorithm and general population-lipid algorithm. This is included in the Discussion section of the paper, second paragraph of 4.1. Given the small differences between the algorithms in performance we felt any additional details might exaggerate the findings.

8. Minor typographical errors which would be eliminated through thorough proof reading.

We thank the reviewer for noting this and have proof read the article again to eliminate typographical errors.

9. The abstract summarises the study well although the focus here is on the economic element despite the title referring to effectiveness and cost-effectiveness of the algorithms. The authors refer to the model as an economical model but it seems to me to be more given work compares the effectiveness and cost-effectiveness of the algorithms.

We note that our model has been referred to as an economic model in our manuscript. Economic evaluations are typically measured via cost-effectiveness which is calculated as the difference in cost divided by the difference in outcome, where outcome may be quality adjusted life years or events. As

outcomes are necessary to calculate cost-effectiveness we thought it was necessary to include the effectiveness outcomes in the manuscript as well.

Reviewer 2 stated:

1. Line 26: The sentence specifies that most algorithms use a cut off of 7.5% or 10%, however to clarify the authors should expand to say that 7.5% risk of developing CVD within 10 years etc... We thank the reviewer for noting the duration of CVD risk for CVD risk thresholds was not specified and have now updated the sentence accordingly.

2. The aim should make clear that the study is looking at both primary and secondary CVD. Some discussion on whether the algorithms chosen are appropriate for use with secondary CVD would be useful.

The CVD risk algorithms employed at baseline to categorise patients into high risk or low risk groups was applied to primary care patients, thus free of existing CVD. Our study did not assess the effectiveness of these algorithms in secondary prevention patients.

The study, did however, include secondary events. A secondary CVD event could only occur if a patient in the model had already had a primary event. Prevention of primary events may have resulted indirectly in the prevention of secondary events.

In summary, secondary events are only included in the model to capture costs and consequences of CVD over 10 years. To comment further on secondary events would suggest a level of precision in the model that we are not confident in providing. They are instead born out in increased costs, reduction in quality of life and risk of mortality.

3. Since the study is set in UK primary care, some description of the primary care context is warranted. For example, which CVD risk scores are recommended for use in primary care, whether these are used in SMI populations. I.e. Are people with SMI eligible for the annual health check program which is supposed to capture QRISK once a year.

We have now included a description of primary care in the UK in the Background section of our manuscript, including the CVD risk score recommended for use in primary care in the UK (QRISK2), as well as information regarding current QOF indicators (annual health checks of blood pressure, smoking and alcohol use for people with SMI as well as the calculation of CVD risk).

4. Line 51-54: It would be useful for the authors to discuss any existing literature around whether PRIMROSE has been found to perform better or worse in SMI populations than other risk scores (if this paper is the first to examine this, then this theme should be brought out earlier in the paper) The Background section of the manuscript references a paper (Osborn DP, Hardoon S, Omar RZ, et al. Cardiovascular risk prediction models for people with severe mental illness: Results from the Prediction and Management of Cardiovascular Risk in People with Severe Mental Illnesses (PRIMROSE) Research Program. JAMA Psychiatry 2015; 72: 143-151) that has previously examined the performance of the PRIMROSE models, compared to general population based models at predicting new CVD events in SMI populations. In the background, we mention that both PRIMROSE models have been shown to perform better at predicting new CVD events than the general population based Framingham algorithms. As the PRIMROSE algorithm has only been developed relatively recently, this is currently the only evidence regarding the performance of this algorithm in SMI populations.

5. The methodology needs to state earlier on which conditions are used to define CVD- I know this

appears later in the description of the simulation models on page 11, but this is worth stating in the opening where the other variables are described.

The Population section of the methodology now includes the cardiovascular conditions used to exclude patients in THIN to ensure our model cohort was primary prevention patients, free of CVD. It also highlights the section to refer to for determination of transition probabilities for primary CVD events and all-cause mortality. The CVD conditions included to determine risk of CVD events in the model were the same conditions patients were excluded with at baseline however to minimise confusion these will only be mentioned in the Transition probabilities section of the manuscript.

6. The methodology should also reference/signpost how SMI and CVD were defined (QOF definition, other?)- which code lists, algorithms were used and how can these be found in order to replicate the disease definitions in other settings (either from other published studies, or to an appendix/online resource).

The SMI definition used to define our population in THIN is included in the Methodology: Population section of the manuscript. SMI was defined as patients with a recorded diagnosis of schizophrenia, bipolar disorder and other long term psychotic illness and/or were listed on the SMI register, in line with our previous study (Hardoon S, Hayes JF, Blackburn R, et al. Recording of severe mental illness in United Kingdom primary care, 2000-2010. PLoS One 2013; 8: e82365). This study demonstrated the rates of SMI in THIN was comparable to incidence rates from previous epidemiological studies of SMI.

The CVD definition has now also been included in the Methodology: Population section of the manuscript. As the population was free of CVD at baseline, CVD was defined as a recorded diagnosis of myocardial infarction, angina pectoris, coronary heart disease, major coronary surgery and revascularization, or cerebrovascular disease including cerebrovascular accident and transient ischaemic attack.

7. The authors state they identified fatal CVD- was this identified from the primary care record or from linked HES/ONS? If so, was linkage complete for all included patients?

The proportion of fatal CVD events in patients who had a primary CVD event was equal to the proportion the patients that had a primary fatal CVD event in the THIN dataset as mentioned in the Transition probabilities section of the manuscript. THIN is linked with HES data. To the best of my knowledge, at the time of analysis, linkage was complete for all included SMI patients in my dataset.

8. The study used 1000 patients due to computational limitations, it would be useful to discuss how the use of more patients/complete patient population would have changed the results. What biases may have been potentially introduced by using a smaller sample.

Before employing a smaller population of 1000 patients in our model, we did multiple testing using larger cohorts of patients to ensure that this smaller subset, whilst computationally feasible, was a true representation and the results were comparable. Whilst the population characteristics were comparable to the larger cohort (Table 1), internal validation of comparing the number of CVD events recorded in THIN and the number of CVD events predicted when no algorithm was employed (baseline), demonstrated similar results, see Strength and limitations section of the discussion. Therefore, using the larger cohort, may change the results marginally, however much of the difference would be attributable to the use of random numbers.

9. Some discussion on how implementation of the SMI-BMI algorithm might be incorporated into current primary care practice/guidelines would give this paper important context. For example, what would the benefits be (if any) over and above existing practices, given that CVD risk assessment is already a part of standard clinical practice and incentivised by QOF. What factors would help practitioners decide whether to invest the time/training in using the SMI specific BMI algorithm.

The benefit of the SMI-specific BMI algorithm, over existing CVD risk algorithms used in current practice in the UK, is that it requires no blood test. This is a limitation of the existing CVD risk algorithms as many people decline blood tests. The ease of use of the SMI-specific BMI algorithm also means that implementation is not limited to primary care settings but it could also be used in mental health settings and non-clinical settings. This would enable more patients with SMI to be targeted for CVD risk assessment and potential CVD risk management. This is described in the Discussion section of the manuscript.

10. The authors stated that they couldn't compare against the most widely used risk scores due to lack of variable availability. Would it have been possible to explore these algorithms using multiple imputation for the missing variables- in order to give a comparison to risks scores already being used in current practice? This would be of great interest to clinical practitioners, for whom this study should help in their decision to move from current practice to potentially adopting a new SMI population specific score.

The THIN dataset which was used to source our patient population did have missing variables, however multiple imputation techniques were used to form a complete dataset for our model population.

The most widely used risk score in the UK in primary care is currently Q-RISK. Whilst the risk score variables are known and risk can be calculated using an online source, the co-efficients for the variables are not available. This means that risk cannot be calculated using an algorithm within the THIN database and would have to be calculated manually for each person in the model. This would have been far too labour intensive given the large numbers involved, particularly when testing reproducibility of results in different samples of our population.

11. Some wider discussion on the challenges of CVD Risk management in SMI populations would strengthen the paper. For example, differences in attendance, adherence, opportunities to identify CVD risk factors may be more important in managing CVD risk than a tailored risk algorithm. In the Discussion section of the manuscript, challenges in CVD risk management in those with SMI has now been included and the opportunities the SMI-specific BMI algorithm presents. We have also included the potential to target people for CVD risk management using other methods such as CVD risk factors as per our current study (Osborn D, Burton A, Walters K, et al. Evaluating the clinical and cost effectiveness of a behaviour change intervention for lowering cardiovascular disease risk for people with severe mental illnesses in primary care (PRIMROSE study): study protocol for a cluster randomised controlled trial. *Trials* 2016; 17: 80).

12. Some discussion on further work would be useful, would an observational cohort study/trial exploring the PRIMROSE risk scores in practice be feasible/useful?

Future work, specifically re-evaluation of the SMI-specific BMI model using real life data after implementation, has now been included in the Conclusions section of the manuscript.

VERSION 2 – REVIEW

REVIEWER	Seren Haf Roberts Bangor University, Wales UK
REVIEW RETURNED	26-Jun-2017

GENERAL COMMENTS	Overall, this is an interesting and important paper demonstrating the
---

	applicability of a CVD risk model for individuals with SMI which does not require a blood tests. This has the potential for improving the way CVD risk is assessed and ultimately managed. I am satisfied that the team have responded appropriately to the comments and suggestions raised in the previous review.
--	---

REVIEWER	Rohini Mathur LSHTM, UK
REVIEW RETURNED	27-Jun-2017

GENERAL COMMENTS	I thank the authors for their comprehensive response to the comments. The paper is greatly improved with the revisions and clarifications suggested- In particular, the methodology is much easier to follow. The rationale for the analysis has been strengthened, and the discussion is better balanced and persuasive. I think the value of highlighting the need for CVD management amongst people with SMI is well made and of relevance to the readership.
--